# Unsolved Issues in Thymic Epithelial Tumour Stage Classification: The Role of Tumour Dimension

**DOI:** 10.3390/diagnostics13223468

**Published:** 2023-11-17

**Authors:** Carolina Sassorossi, Pietro Bertoglio, Filippo Lococo, Gloria Santoro, Elisa Meacci, Dania Nachira, Maria Teresa Congedo, Jury Brandolini, Matteo Petroncini, Adriana Nocera, Diepriye Charles-Davies, Piergiorgio Solli, Stefano Margaritora, Marco Chiappetta

**Affiliations:** 1UOC di Chirurgia Toracica, Università Cattolica del Sacro Cuore, 00168 Rome, Italy; marco.chiappetta@policlinicogemelli.it (M.C.); filippo.lococo@policlinicogemelli.it (F.L.); elisa.meacci@unicatt.it (E.M.); dania.nachira@policlinicogemelli.it (D.N.); mariateresa.congedo@policlinicogemelli.it (M.T.C.); adriana.nocera91@gmail.com (A.N.); stefano.margaritora@policlinicogemelli.it (S.M.); 2UOC di Chirurgia Toracica, Fondazione Policlinico Universitario A. Gemelli IRCCS, 00168 Rome, Italy; 3Divisione di Chirurgia Toracica IRCCS Azienda Ospedaliera Universitaria Bologna, 40138 Bologna, Italy; pieberto@hotmail.com (P.B.); jury.brandolini@ausl.bologna.it (J.B.); matteo.petroncini@gmail.com (M.P.); piergiorgio.solli@gmail.com (P.S.); 4Chirurgia Toracica, Alma Mater Studiorum, Università di Bologna, 40126 Bologna, Italy; 5UOC di Chirurgia Generale, Fondazione Policlinico Universitario A. Gemelli IRCCS, 00168 Rome, Italy; gloria.santoro@policlinicogemelli.it; 6Dipartimento di Diagnostica per Immagini, Radioterapia Oncologica ed Ematologia, Fondazione Policlinico Agostino Gemelli, 00168 Roma, Italy; charlesdavies.diepriye@guest.policlinicogemelli.it

**Keywords:** thymic epithelial tumour, staging system, surgery, oncology, TNM

## Abstract

According to the different classifications now in use, thymic tumours are staged by the extent of local invasiveness, and tumour size is not included as a major determinant for the T category. The aim of this double-site retrospective study is to analyse the correlation between tumour dimension and overall survival (OS) in patients who underwent surgical treatment. From January 2000 to December 2020, patients with thymic epithelial tumours who underwent surgical resection were included in this study. Data from a total of 332 patients were analysed. Five- and ten-year overall survival (5–10 YOS) was 89.26% and 87.08%, respectively, while five- and ten-year disease-free survival (DFS) was 88.12% and 84.2%, respectively. Univariate analysis showed a significant correlation between male sex (*p*-value 0.02), older age (*p*-value < 0.01), absence of myasthenia gravis (*p*-value < 0.01), increase in pTNM (pathological Tumor Node Metastasis) (*p*-value 0.03) and increase in the number of infiltrated organs (*p*-value 0.02) with an increase in tumour dimension. Tumour dimension alone was not effective in the prediction of DFS and OS, both when considered as a continuous variable and when considered with a cut-off of 3 and 5 cm. However, with multivariate analysis, it was effective in predicting OS in the aforementioned conditions (*p*-value < 0.01). Moreover, multivariate analysis was also used in the thymoma and Masaoka I subgroups. In our experience, the role of tumour dimension as a descriptor of the T parameter of the TNM (Tumor Node Metastasis) staging system seemed to be useful in improving this system.

## 1. Introduction

Thymic epithelial neoplasms are rare organ-specific neoplasms with varying malignant potential that comprise thymomas, thymic carcinomas (TC) and thymic neuroendocrine tumours (NETT).

According to the different existing classifications, particularly in the Masaoka–Koga and TNM 8th edition, thymic tumours are staged by the extent of local invasiveness, but size is not considered as a major determinant for the T category [1,2,3].

The Masaoka staging system for thymoma was presented in 1981 and has been accepted as the global standard staging system for more than 20 years [4]. In 2010, the International Thymic Malignancy Interest Group (ITMIG) was established with the purpose of developing a new staging system, eventually proposed in 2016. It was a TNM classification system, included in the 8th TNM staging classification. In this novel staging, T was defined by the level of invaded adjacent organs (close to the Masaoka staging system) without taking into account tumour dimension.

In 2022, The International Association for the Study of Lung Cancer Thymic Epithelial Tumour Staging Project [1] described the unsolved issues identified in this current stage classification, which are worth addressing and discussing for the ninth edition of the TNM classification proposed for 2024. Among these issues, the prognostic relevance of tumour size for thymoma has been pointed out to be discussed and clarified.

In the making of the 8th edition of the Union for International Cancer Control (UICC)/American Joint Committee on Cancer (AJCC) TNM classification in 2014, in the International Association for the Study of Lung Cancer (IASLC)/International Thymic Malignancy Interest Group (ITMIG) Thymic Epithelial Tumours Staging Project, a database of 5796 patients was analysed and considered one-dimension measurement of the tumour. The analysis showed that tumour size at a 10 cm threshold was only significant in incomplete (R1 and R2) resection and advanced (stages III and IV) tumours [5]. As a result, tumour size was not considered for the eighth edition of the TNM classification of thymic epithelial tumours.

Accordingly, the first issue to be considered is the prognostic role of tumour size itself. To date, no definitive or clear results have revealed the predictive value of tumour size for thymoma prognosis [4,6,7,8].

Nicholson et al. [5] did not find survival differences according to the tumour size in patients with completely resected thymic tumours or in an early stage. Similarly, a retrospective study by Ruffini et al. that included more than 2000 patients from the ESTS database identified that tumour size was neither a predictor of overall survival (OS) nor of recurrent-free survival [8,9]. Other studies found a role of tumour dimension in prognosis, but they all reported different thresholds for tumour size. Indeed, for example, Fukui et al. [7] indicated a better relapse-free survival when the tumour size was less than 4 cm on the longest diameter of the specimen. The Japanese Association for Research of the Thymus [4] analysed a national database of 2083 treated cases nationwide and concluded that tumour size was an independent determinant of RFS and disease-specific survival with a cut-off of 5 cm and 8 cm, respectively.

Moreover, much clarity is still needed on the role of tumour dimension in prognosis, either as an independent risk factor or if it depends on the association with other tumour characteristics (stage, completeness of resection, histology, etc.). In the IASLC/ITMIG Thymic Epithelial Tumours Staging Project, the 10 cm threshold only had a role in the R1 and R2 tumours as well as in advanced stages (III e IV). Fukui and colleagues [7], who defined the 4 cm threshold, also confirmed the fact that tumour dimension better defines the prognosis for thymic carcinomas (TC) and thymic neuroendocrine tumours (NETT).

The aim of this study is to analyse the correlation between tumour dimension and OS in patients who underwent surgical treatment.

## 2. Materials and Methods

Patients with thymic epithelial tumours who underwent surgical resection were included in this study. Data from a total of 332 patients were analysed.

The pre-operative assessment included a contrasted CT scan and magnetic resonance if indicated in the case of suspected infiltration of neighbouring organs.

Thymic epithelial tumour was suspected on the basis of radiological characteristics, and a confirmatory biopsy was performed in case of doubt or in the case of patients who may benefit from neoadjuvant treatment.

Surgery was performed with general anaesthesia. All patients underwent neurological examination before surgery to investigate the presence of myasthenia gravis. The analysis included the anti-acetylcholine receptor, anti-Mu SK antibodies and electromyography (EMG).

Surgery was performed with median sternotomy or the combination of cervicectomy and manubrial section. After the introduction of minimally invasive techniques, in addition, video-assisted thoracic surgery (VATS) and robotic resection were performed in selected cases, mainly for tumours smaller than 3 cm. All the operations were performed with the aim of radical resection. In the case of suspected infiltrations, where possible, infiltrated structures were removed in bloc with the tumour, and the area of removal was signed with titanium clips.

The pathological specimen was analysed and staged with the WHO histological classification [3] and Masaoka staging system [2]. After 2014, the TNM classification was taken into account and considered for staging. For patients operated on before 2014, pathological reports were reconsidered, and the TNM stage was added.

Induction therapy was administrated considering pre-operative imaging, while adjuvant therapy was considered according to the pathological results. The choice of radiotherapy or chemotherapy depended on the pathological stage, patient characteristics and previous treatment. After surgery, follow-up was conducted according to the most recent available guidelines. For instance, a baseline thoracic CT scan was carried out 3 months after surgery. For completely resected stage I/II thymomas, CT scans were performed every year for 5 years and then every 2 years. For stage III/IV thymomas, thymic carcinoma or after R1–2 resection, CT scans were performed every 6 months for 2 years and then annually. Follow-up was continued for 10–15 years [10,11].

The primary endpoint was the following:Overall survival (OS).

Secondary endpoints were the following:Disease free survival (DFS)Correlation between tumour dimension and other pathologic characteristics.

Tumour dimension was studied as a continuous variable with two different cut-offs at 3 cm and 5 cm. The variables taken into account were sex, age, presence of myasthenia gravis, kind of surgery, pTNM, Masaoka stage, histology, number of infiltrated organs and recurrence. Through univariate and multivariate analyses, we investigated the prognostic value of tumour size for thymic epithelial tumour (TET) according to tumour stage after complete resection.

### Statistical Analysis

Continuous variables were analysed using the mean and standard deviations (SD) and compared using Welch’s *t* test, based on the unequal population variance. In addition, the ANOVA test and non-parametric Kruskal–Wallis test were considered to better explore the association of the features to our conditions of interest, whereas categorical variables were analysed using frequencies and percentages and were compared using the chi-square test. Based on 11% of not available data (NA) for tumour dimension—cm (outcome of interest), imputation with the *mice* package in R 4.1.2 was performed to obtain a full and complete dataset.

Therefore, survival analyses were conducted using the Kaplan–Meier method with the log-rank test. Secondly, a Cox proportional-regression model was performed, considering a cluster model based on the referred Surgical Centre, due to the aim of our study. Overall survival (OS) was calculated from the date of surgery to death for any cause, while DFS was calculated from the time of surgery to the first detection of recurrence. The statistical analysis and the graphical related part were performed with software R 4.1.2, implementing with the libraries *MASS*, *survival*, *survminer* and *ggplot2* downloaded from CRAN (https://cran.r-project.org/, accessed on 10 August 2023).

## 3. Results

### 3.1. Overall

The main clinical and pathological characteristics are reported in Table 1. The mean age at surgery was 57.1 ± 15.4 yrs. Myasthenia gravis affected 50.6% of patients. Resection was complete (R0) in 96.9% of the cases. The most frequent Masaoka stage was stage 2 (57.9%), while the most common pTNM stage was stage pt1 (86.7%). The mean tumour dimension was 5.12 ± 2.91 cm. The five- and ten-year overall survival (5–10 YOS) was 89.26% and 87.08%, respectively, whereas the five- and ten-year DFS was 88.12% and 84.2%, respectively.

### 3.2. Tumour Dimension and Association with Variables

All the variables analysed in the univariate models are reported in Table 2. Univariate analysis indicated a significant association between male sex (*p*-value 0.02), older age (*p*-value < 0.01), absence of myasthenia gravis (*p*-value < 0.01), considering non-parametric test with the increase in TNM (*p*-value 0.03) and increase in the number of infiltrated organs (*p*-value 0.02) with the increase in tumour dimension. In particular, regarding the number of infiltrated organs, it was significant if the number was 3 vs. 0 and 3 vs. 1, considering a Tukey HSD test with family-wise error rate (FWER) correction.

On the contrary, there was no statistical correlation between the histology (thymoma vs. carcinoma), the status of the capsule (infiltrated or not) and the tumour dimension.

### 3.3. Survival Outcomes

Regarding the Masaoka–Koga staging system, both the DFS (*p*-value 0.0019) and OS (*p*-value 0.021) for stage I and II curves overlapped, while a greater difference was evident for stage III and stage IV (Appendix A).

With reference to the number of infiltrated organs, a clear stratification for both DFS (*p*-value 0.0083) and OS (*p*-value 0.03) was observed when the number of infiltrated organs was three compared to one and zero (Appendix A).

The TNM staging system was associated only with the prediction of DFS (*p*-value 0.014), while it was not statistically related to the OS (Appendix A). Cox regression analysis was performed for DFS and OS according to Masaoka–Koga, pTNM and the number of infiltrated organs. The number of infiltrated organs was effective in predicting DFS (*p*-value < 0.01) with an increasing hazard ratio of recurrence (HR) for one (HR: 3.86), two (HR: 4.87) and three (HR: 12.56) infiltrated organs.

This variable was also effective in predicting the OS (*p*-value 0.05 for one infiltrated organ and *p*-value < 0.01 for three infiltrated organs) with a stratification for the number of organs infiltrated, decreasing HR and the number increase (HR was approximately 0 for three organs infiltrated).

Tumour dimension was not effective in the prediction of DFS and OS, both when considered as a continuous variable and when considered with the cut-off of 3 (Figure 1) and 5 cm (Figure 2).

### 3.4. Multivariable Models for Disease-Free Survival and Overall Survival

Tumour dimension, considered as a continuous variable with the cut-off at 3 cm and 5 cm, was involved in the multivariate analysis, which took into account the number of infiltrated organs and the pTNM stage.

For DFS, either as continuous or with the cut-off of 3 and 5 cm, tumour dimension was found to be significant (Appendix A).

On the contrary, tumour dimension was effective in the prediction of OS when considered as a continuous variable (*p*-value < 0.01) with the cut-off of 3 cm (*p*-value < 0.01) and 5 cm (*p*-value < 0.01) (Table 3).

Considering only thymoma-affected patients in the multivariate models, a cut-off of 5 cm was important for the prediction of DFS (*p*-value < 0.01) (Appendix A). As a continuous variable (*p*-value < 0.01) with the cut-off of 3 (*p*-value < 0.01) and 5 cm (*p*-value 0.01), tumour dimension was involved in the prediction of OS (Appendix A).

With regards to only Masaoka I patients in the multivariate models, tumour dimension with cut-offs set at 3 or 5 cm as a continuous variable was capable of predicting DFS (*p*-value < 0.01) (Table 4). However, it was not effective in predicting OS for the three conditions mentioned above (Appendix A).

## 4. Discussion

In our study, we investigated the survival outcomes for TET in a large two-centred cohort with a particular interest on the tumour dimension.

In general, according to previous studies [12,13,14], we found that the completeness of the Masaoka–Koga staging system, pTNM staging system and number of infiltrated organs [14] is an independent prognostic factor for TETs. As for tumour size, most of the previous studies have analysed tumour size as a secondary covariate to correct other major factors in the multivariate [13].

In our experience, tumour size was first analysed as an independent factor in the definition of OS and DFS. We considered the dimension with the cut-offs of 3 and 5 cm as a continuous variable. For both DFS and OS, the tumour dimension was not significantly associated with survival outcomes. In contrast, in the K–M curves, we observed the tendency of the *p*-value to be significant when the cut-off was set at 3 cm (*p* = 0.086) for the OS. This observation was much more evident for the OS when the variable was continuous, even with a *p*-value of >0.05.

Based on the results obtained in the univariate analysis for the Masaoka–Koga staging system in relation to the number of infiltrated organs and tumour dimension, a multivariate analysis was performed with the Cox regression model. In this case, as a covariate variable continuous with the cut-off of 3 and 5 cm, the tumour dimension was not involved in the prediction of DFS. On the other hand, more encouraging results were obtained with OS. Indeed, in the Cox regression model, the tumour dimension was significant in predicting the OS for the three conditions mentioned above (continuous variable, cut-off of 3 and 5 cm) (*p* < 0.01).

Similarly, we performed the same multivariate analysis only in thymoma cases and then in Masaoka I-affected patients. Indeed, the IASLC group of study for the ninth edition of the TNM (1) suspects that the role of tumour size may be different according to the histology and to the level of local invasiveness. In the thymoma sub-group, we found no statistically significant correlation between DFS and tumour dimension set as a continuous variable with the cut-off of 3 cm. However, the tumour dimension with the cut-off set at 5 cm was significant in the prediction of DFS (*p* < 0.01). Furthermore, the tumour dimension was significantly associated with OS in all conditions investigated (*p* < 0.01). In the Masaoka I sub-group, the tumour dimension was not predictive of OS, even when set as a continuous variable with the cut-off of either 3 cm or 5 cm. Instead, the three considered conditions were predictive for DFS (<0.01).

According to the outcomes of previously published studies, tumour size is a prognostic factor of recurrence or survival. Moreover, the cut-off values of tumour sizes vary with reported thresholds from 4 to 11 cm [3,15].

Ruffini and colleagues [16] investigated the role of the tumour dimension and reported that an increased tumour size was not a significant predictor of either OS or DFS. Although, as an adjusted variable, it was a predictor of incomplete resection. Smaller tumours were generally found to be associated with improved survival and decreased risk of recurrence [17]. In addition, a review published before 2011 reported that tumour dimension, generically defined as small vs. large tumour, was found to be significant in only 36% of the studies for OS and 43% for DFS [18].

The current TNM staging system is based on the infiltration level. Pericardial infiltration is staged as T2, while other infiltrated structures are staged as T3. This anatomical difference did not reflect a significant difference in OS [5]. Therefore, the actual TNM staging system still does not reflect a difference in survival outcomes. On the other hand, our findings are encouraging. In the multivariate analysis, the dimension was an effective factor predicting OS as a continuous variable with the cut-offs of 3 and 5 cm when the number of infiltrated organs were one or three. When we considered only the thymoma-affected patients, the cut-off of 5 cm was also effective for the prediction of DFS. These findings are in line with the recent work of Sakai and colleagues [19] who found the cut-off of 5 cm to be prognostic for OS in their multivariate analysis, together with the actual TNM stage.

In our opinion, the tumour dimension itself is not a parameter robust enough to define the survival outcomes and the T descriptor alone. According to our results, the tumour dimension can be predictive of OS when associated with other parameters, especially with the number of infiltrated organs (three vs. one) (Table 4). Consequently, it should be taken into consideration that the T descriptor of the TNM staging system could be a composite parameter with the number of infiltrated organs.

Concerning the definition of a cut-off to better stratify the risk, we considered 3 and 5 cm. Alone, none of the cut-offs was effective in the prediction of OS and DFS. However, the results from the multivariate analysis were encouraging, as these cut-offs were found to be significant for the prediction of OS (in association with the number of infiltrated organs).

Our study presents some limitations. First, our study is limited by its retrospective nature. Second, our analysis was performed on a limited number of patients. Although our results are encouraging, further studies with a larger cohort are needed to validate our findings. Tumour dimension may be used to describe the T factor of the TNM staging system to improve survival stratification in these patients. Furthermore, the optimal cut-off value for tumour size has to be defined to distinguish different prognoses in completely resected patients.

## 5. Conclusions

In our experience, the role of tumour dimension as a descriptor of the T parameter of the pTNM staging system seemed to be a potential factor to improve this system, as it was related to OS and DFS in the multivariate analysis, together with the number of infiltrated organs. Further evaluations in larger cohorts are necessary to define the dimension cut-off for T stratification and the eventual introduction of a composite T parameter.

Furthermore, our study confirms the effectiveness of the Masaoka–Koga staging system and number of infiltrated organs for the prediction of OS, and the Masaoka–Koga staging system, number of infiltrated organs and TNM staging system for the prediction of tumour recurrence.

## Figures and Tables

**Figure 1 diagnostics-13-03468-f001:**
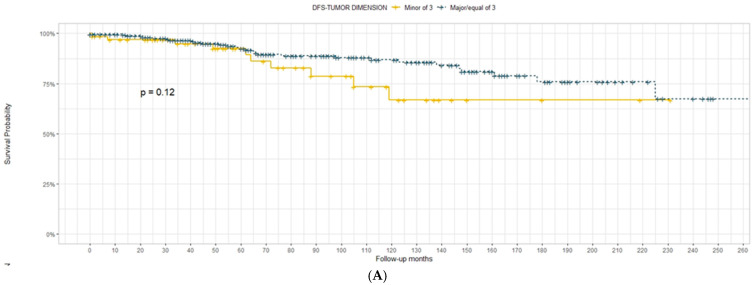
Disease-free survival (**A**) and overall survival (**B**) according to tumour dimension with 3 cm cut-off.

**Figure 2 diagnostics-13-03468-f002:**
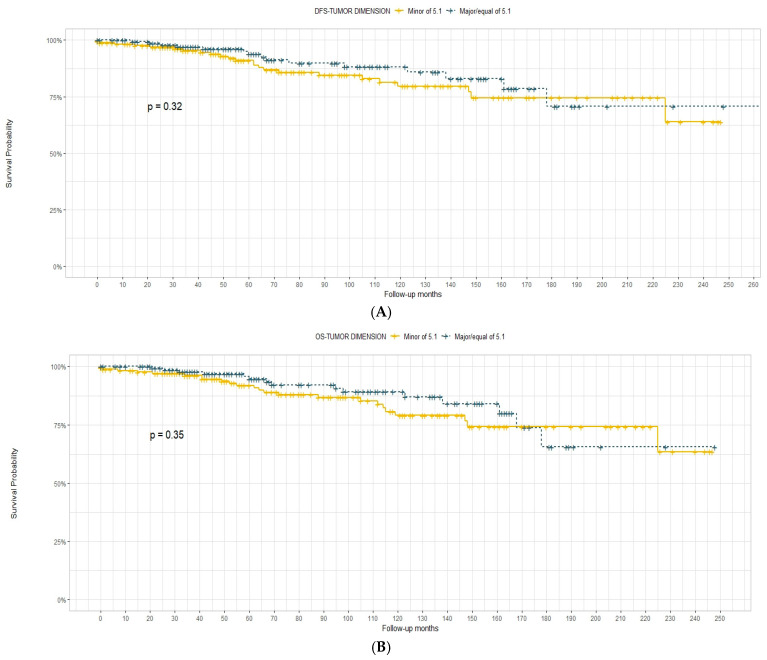
Disease-free survival (**A**) and overall survival (**B**) according to tumour dimension with 5 cm cut-off.

**Table 1 diagnostics-13-03468-t001:** Main clinical and pathological characteristics.

Variables	Description
Sex—*n* (%)	
Female	189 (56.9%)
Male	143 (43.1%)
Age (at surgery)	
Mean ± SD	57.1 ± 15.4
Median [Q1–Q3]	4.5 [3.0–6.5]
Tumour dimension (cm)	
Mean ± SD	5.12 (2.91)
Median [Q1–Q3]	4.5 [3.0–6.5]
Number of infiltrated organs—*n* (%)	
0	283 (85.2%)
1	28 (8.5%)
2	17 (5.1%)
3	3 (1.2%)
Masaoka—*n* (%)	
I	96 (29.3%)
II	190 (57.9%)
III	33 (10.1%)
IV	9 (2.7%)
Myasthenia gravis—*n* (%)	
No	162 (49.4%)
Yes	166 (50.6%)
pTNM—*n* (%)	
pt1	287 (86.7%)
pt2	31 (9.4%)
pt3	13 (3.9%)
Disease Residue—*n* (%)	
R0	321 (96.9%)
R1	10 (3.1%)
Thymoma or Carcinoma—*n* (%)	
Carcinoma	17 (5.1%)
Thymoma	315 (94.9%)
Histological subtypes of Thymoma—*n* (%)	
A AB B1 B2 B3 Mixed (B1/B2, B2/B3, B1/B3)	19 (5.7%) 85 (25.6%) 46 (13.8%) 107 (32.4%) 27 (8.1%) 31 (9.4%)
Low or High Grade—*n* (%)	
Low Grade	268 (80.7%)
High Grade	64 (19.3%)
NAD—*n* (%)	
No NAD Therapy	201 (61.7%)
Chemotherapy	7 (2.2%)
Radiotherapy	112 (34.3%)
Radiochemotherapy	6 (1.8%)
Capsule Microscopic Infiltration—*n* (%)	
No Infiltration	93 (28.0%)
Only Capsule	85 (25.6%)
Capsule and Fat	154 (46.4%)

**Table 2 diagnostics-13-03468-t002:** Univariate analysis.

Tumour Dimension (cm) vs. Other Characteristics			
	Levels	Mean (sd)	*p*-Values
Sex	Female	4.8 (2.7)	0.02 *
	Male	5.5 (3.1)	
Age (at surgery)	[14.0, 120.0]	5.1 (2.9)	<0.01 *
Number of infiltrated organs	0	5.0 (2.9)	(3 vs. 0) 0.03 ^$^
	1	4.9 (3.0)	(3 vs. 1) 0.02 ^$^
	2	6.1 (2.1)	
	3	9.0 (3.7)	
Masaoka	[1.0, 4.0]	5.1 (2.9)	0.35
Capsule Microscopic Infiltration	Capsule and Fat	5.2 (2.7)	0.58
	NO	5.1 (3.1)	
	Only Capsule	5.1 (3.1)	
Myasthenia	NO	5.6 (3)	<0.01 *
	YES	4.7 (2.7)	
pTNM	pt1	4.9 (2.8)	0.03 **
	pt2	6.6 (3.1)	
	pt3	5.7 (3.1)	
Disease Residue	R0	5.1 (2.9)	0.25
	R1	5.5 (3.8)	
Organs Infiltrated	NO	5.0 (2.9)	0.33
	YES	5.6 (3.1)	
Thymoma VS. Carcinoma	Carcinoma	4.6 (1.6)	0.99
	Thymoma	5.2 (2.9)	
Low High Grade	High Grade	5.4 (3.2)	0.23
	Low Grade	4.9 (2.8)	
NAD	Chemotherapy	5.8 (4.5)	0.59
	No NAD	5.0 (3.0)	
	Radiotherapy	5.2 (2.8)	
	Radio/Chemotherapy	5.5 (2.5)	

* Welch two sample *t*-test; ** Kruskall–Wallis test; ^$^ ANOVA test, *p*-value adjusted (FWER).

**Table 3 diagnostics-13-03468-t003:** Multivariate model for overall survival.

	Coef	Exp (Coef)	Lower 95% CI	Upper 95% CI	SE (Coef)	Robust-SE	z-Test	Pr(>|z|)
Continuous variable	−1.25 × 10^−1^	8.83 × 10^−1^	8.27 × 10^−1^	9.43 × 10^−1^	7.27 × 10^−2^	3.35 × 10^−2^	−3.73	<0.01
Number of Organs Infiltrated: 1	8.78 × 10^−1^	2.41	1.35	4.30	5.18 × 10^−1^	2.96 × 10^−1^	2.97	<0.01
Number of Organs Infiltrated: 2	4.83 × 10^−1^	1.62	8.12 × 10^−1^	3.24	6.69 × 10^−1^	3.53 × 10^−1^	1.36	0.17
Number of Organs Infiltrated: 3	−1.52 × 10^1^	2.61 × 10^−7^	6.07 × 10^−8^	1.12 × 10^−6^	4.04 × 10^−3^	7.44 × 10^−1^	−20.38	<0.01
Masaoka	2.87 × 10^−1^	1.33	6.18 × 10^−1^	2.87	2.95 × 10^−1^	3.92 × 10^−1^	0.73	0.46
Cut-off 5 cm	−2.90 × 10^−1^	7.48 × 10^−1^	7.09 × 10^−1^	7.90 × 10^−1^	3.35 × 10^−1^	2.79 × 10^−2^	−10.41	<0.01
Number of Organs Infiltrated: 1	8.77 × 10^−1^	2.41	1.33	4.35	5.26 × 10^−1^	3.02 × 10^−1^	2.90	<0.01
Number of Organs Infiltrated: 2	4.84 × 10^−1^	1.62	7.69 × 10^−1^	3.43	6.72 × 10^−1^	3.81 × 10^−1^	1.27	0.20
Number of Organs Infiltrated: 3	−15.40	2.01 × 10^−7^	4.29 × 10^−8^	9.42 × 10^−7^	4.07 × 10^−3^	7.88 × 10^−1^	−19.56	<0.01
Masaoka	2.60 × 10^−1^	1.30	6.00 × 10^−1^	2.81	2.99 × 10^−1^	3.94 × 10^−1^	0.66	0.51
Cut-off 3 cm	−7.44 × 10^−1^	4.75 × 10^−1^	3.66 × 10^−1^	6.17 × 10^−1^	3.84 × 10^−1^	1.34 × 10^−1^	−5.57	<0.01
Number of Organs Infiltrated: 1	7.54 × 10^−1^	2.13	1.14	3.98	5.27 × 10^−1^	3.20 × 10^−1^	2.36	0.01
Number of Organs Infiltrated: 2	4.62 × 10^−1^	1.59	7.03 × 10^−1^	3.58	6.82 × 10^−1^	4.16 × 10^−1^	1.11	0.26
Number of Organs Infiltrated: 3	−1.56	1.65 × 10^−7^	3.50 × 10^−8^	7.78 × 10^−7^	4.12 × 10^−3^	7.91 × 10^−1^	−19.73	<0.01
Masaoka	3.41 × 10^−1^	1.41	6.20 × 10^−1^	3.19	3.06 × 10^−1^	4.18 × 10^−1^	0.81	0.41

**Table 4 diagnostics-13-03468-t004:** Multivariate model for disease-free survival in Masaoka I.

	Coef	Exp (Coef)	Lower 95% CI	Upper 95% CI	SE (Coef)	Robust-SE	z-Test	Pr(>|z|)
Continuous variable	3.23 × 10^−1^	1.38	1.31	1.45	1.28 × 10^−1^	2.55 × 10^−2^	12.62	<0.01
Number of Organs Infiltrated: 1	−1.28 × 10^1^	2.59 × 10^−6^	2.44 × 10^−7^	2.75 × 10^−5^	1.11 × 10^4^	1.21	−10.67	<0.01
Number of Organs Infiltrated: 2	3.81	4.55 × 10^1^	2.81 × 10^1^	7.39 × 10^1^	1.27	2.47 × 10^−1^	15.46	<0.01
cut-off 5 cm	1.20	3.31	2.52	4.37	1.15	1.40 × 10^−1^	8.54	<0.01
Number of Organs Infiltrated: 1	−1.34 × 10^1^	1.45 × 10^−6^	1.74 × 10^−7^	1.21 × 10^−5^	1.27 × 10^4^	1.08	−12.41	<0.01
Number of Organs Infiltrated: 2	3.89	4.93 × 10^1^	3.86 × 10^1^	6.31 × 10^1^	1.44	1.25 × 10^−1^	31.18	<0.01
cut-off 3 cm	3.83 × 10^7^	3.83 × 10^7^	4.96 × 10^6^	2.96 × 10^8^	5.67 × 10^3^	1.04	16.74	<0.01
Number of Organs Infiltrated: 1	−1.75 × 10^1^	2.47 × 10^−8^	2.80 × 10^−9^	2.19 × 10^−7^	5.82 × 10^4^	1.11	−15.74	<0.01
Number of Organs Infiltrated: 2	2.88	1.79 × 10^1^	1.17 × 10^1^	2.74 × 10^1^	1.15	2.16 × 10^−1^	13.33	<0.01

## Data Availability

Data are available on request due to restrictions, e.g., privacy or ethical. The data presented in this study are available on request from the corresponding author.

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
