# Peer review of "Unsolved Issues in Thymic Epithelial Tumour Stage Classification: The Role of Tumour Dimension"

_diagnostics, 2023, doi:10.3390/diagnostics13223468_

Round 1

Reviewer 1 Report

Comments and Suggestions for Authors

The authors reported an interesting series of 332 thymic epithelial tumors evaluating the importance of size of TET in the prognosis. 

they found that the size was not prognostic on univariate analysis for DFS and OS but was prognostic on OS on multivariate analysis

Major comments

1- It is important in this series to have different multivariate analysis to understand better the results

We need to have the results multivariate analyses  considering

a) the size with the three methodologies and the pTNM

b) the size with the Masaoka Koga

2) When assessing the number of infiltrated organs we need to consider either 3 vs 0, 1, 2 or 3,2 vs 0, 1 or 3,2, 1 vs 0 to evaluate the best prognostic factor on univariate analysis and to include it in a multivariate analysis. However, when evaluating the number of infiltrated organs, we need not to include TNM  or MK in the multivariate which depends on number of infiltrated organs 

Minor comments

1- For table 1 the organs considered in the number of infiltrated organs should be listed as a footnote

2- Masaoka Koga : use I, II, III, IV

3 for TNM , replace by pTNM, pT1, pT2, pT3

4- concerning carcinoma and thymoma: define thymoma subtype as well as carcinoma subtype including neuroendocrine

5 why considering capsule infiltration as not included in any classification?

Comments on the Quality of English Language

many editing mistakes : i.e. page 2 line 87 epitalial; line 91 Miastenia.....

Author Response

Reviewer 1:

The authors reported an interesting series of 332 thymic epithelial tumors evaluating the importance of size of TET in the prognosis.

They found that the size was not prognostic on univariate analysis for DFS and OS but was prognostic on OS on multivariate analysis

Major comments

Comment 1: It is important in this series to have different multivariate analysis to understand better the results. We need to have the results multivariate analyses considering:

  1. a) the size with the three methodologies and the pTNM
  2. b) the size with the Masaoka Koga

Answer: thank you for your observation. We  considered pTNM and Masaoka Koga in some multivariable models, reported in table 3, 4, 5, 6, 7, 8. we think this is an appropriate analysis, but if you have more suggestions of the kind of multivariable models please do not hesitate to ask to us and we will try to provide.

Correction: none.

Comment 2: When assessing the number of infiltrated organs, we need to consider either 3 vs 0, 1, 2 or 3,2 vs 0, 1 or 3,2, 1 vs 0 to evaluate the best prognostic factor on univariate analysis and to include it in a multivariate analysis. However, when evaluating the number of infiltrated organs, we need not to include TNM or MK in the multivariate which depends on number of infiltrated organs.

Answer: Thank you for your observation. Concerning the multivariate analysis, we took into account the number of infiltrated organs, pTNM and Masaoka Koga staging system. Actually, we considered the three of them because the TNM and Masaoka are influenced by the level of organs or structure infiltrated rather than from the number of infiltrated organs. This consideration was also reported in the discussion section (line 277-279) (ref 5). About the comparison of the number of infiltrated organs (3 vs 0,1,2 etc.) we made the correction in the text and we added a table in the supplementary material.

Correction: line 168-170, table 2b

Minor comments

Comment 1: For table 1 the organs considered in the number of infiltrated organs should be listed as a footnote.

Answer: thank you for your nice observation. We added a footnote to list the infiltrated organs considered.

Correction: line 169

Comment 2: Masaoka Koga: use I, II, III, IV

Answer: thank you for your suggestion. We corrected through the text the masaoka koga staging system.

Correction: table 1 and through the text

Comment 3:  for TNM, replace by pTNM, pT1, pT2, pT3

Answer: thank you for your useful suggestion, we corrected it through the text

Correction: table 1, 2, 3, 5 and through the text

Comment 4: concerning carcinoma and thymoma: define thymoma subtype as well as carcinoma subtype including neuroendocrine.

Answer: thank you for your suggestion. We provided the sub classification in table 1. In our series, we had not neuroendocrine thymic tumour.

Correction: table 1

Comment 5: why considering capsule infiltration as not included in any classification?

Answer: thank you for your comment. The capsule infiltration was considered (table 1) but it came out that the capsule infiltration was not statistically associated with the increasing of the tumour dimension. Then, for the multivariate analysis, as we wrote (line 222-224), we took into account the tumour variable that resulted associated with dimension in the univariate (so the pTNM and the number of infiltrated organs). However, we will better specify this issue in the results

Correction: line 187.

Reviewer 2 Report

Comments and Suggestions for Authors

The manuscript by Sassorossi et al. deals with some unsolved issued regarding the stage classification of thymic epithelial neoplasms. It is an interesting study focusing on the possible role of the tumor diameter and/or number of infiltrated organs in tumor prognosis and raises the question of adding these parameters in T-status of TNM classification.

However, I believe there are some things that need to be corrected before publication can be taken into consideration:

1.       There are significant flaws regarding the English language (see title of the manuscript, text and tables that should be corrected accordingly.

2.       The authors documented significant associations of number of infiltrated organs and tumor diameter with other parameters such as prognosis, using several cut-offs (for example for number of infiltrated organs 3vs 0 and 3 vs 1, or for tumor diameter either 3cm or 5cm). A correction for multiple comparisons (such as Bonferroni) is necessary in such cases.

3.       Please modify table 7 in order to be more readable regarding the coefficients and providing only information regarding lower and upper 95% CI and p-value. In addition, in this table, it is not clear how many Cox models are adjusted and the significant results are not emphasized. Please modify accordingly.

4.       In my opinion, there should be a complete revision of the Discussion section in terms of presentation. The authors use sometimes only a sentence as a paragraph and this makes the whole section difficult to follow. Please try to organize your comments upon your results in this section in paragraphs and not in separate sentences.

Comments on the Quality of English Language

Please see my first comment. A revision by an english native speaker is  necessary.

Author Response

Reviewer 2:

The manuscript by Sassorossi et al. deals with some unsolved issued regarding the stage classification of thymic epithelial neoplasms. It is an interesting study focusing on the possible role of the tumour diameter and/or number of infiltrated organs in tumour prognosis and raises the question of adding these parameters in T-status of TNM classification.

However, I believe there are some things that need to be corrected before publication can be taken into consideration:

Comment 1: There are significant flaws regarding the English language (see title of the manuscript, text and tables that should be corrected accordingly.

Answer: thank you for your useful suggestion. We made major language revision.

Correction: Through the text

Comment 2: The authors documented significant associations of number of infiltrated organs and tumour diameter with other parameters such as prognosis, using several cut-offs (for example for number of infiltrated organs 3 vs 0 and 3 vs 1, or for tumour diameter either 3cm or 5cm). A correction for multiple comparisons (such as Bonferroni) is necessary in such cases.

Answer: thank you for your interesting observation. We added a more precise description of how we made the statistical analysis for the number of infiltrated organs.

Correction: line 168-170, table 2b

Comment 3: Please modify table 7 in order to be more readable regarding the coefficients and providing only information regarding lower and upper 95% CI and p-value. In addition, in this table, it is not clear how many Cox models are adjusted and the significant results are not emphasized. Please modify accordingly.

Answer: thank you for your useful observation. We corrected table 7 to make the results clearer, with the significative ones in bold.

We think that it is important to specify all the parameters of the models to better understand the real conceptualization of the statistical significance. The cox models used were for the tumour dimension was analysed as continuous variable, and also as a dichotomous variable based on two up/down cut off of 3 and 5 cm.

Correction: table 7

Comment 4: In my opinion, there should be a complete revision of the Discussion section in terms of presentation. The authors use sometimes only a sentence as a paragraph and this makes the whole section difficult to follow. Please try to organize your comments upon your results in this section in paragraphs and not in separate sentences.

Answer: thank you for your useful suggestion. We tried to re organize the discussion to make it easier to be read.

Correction: discussion section

Round 2

Reviewer 1 Report

Comments and Suggestions for Authors

The authors responded well to all the comments made and modified the manuscript accordingly

Reviewer 2 Report

Comments and Suggestions for Authors

I do not have any suggestions to the Authors. Thank you for taking my comments into consideration.